# Listwise Generalized Preference Optimization with Process-Aware Signals for LLM Reasoning

## Abstract

Standard preference optimization methods for LLMs suffer from two limitations: pairwise objectives like DPO discard valuable ranking information, and outcome-only supervision provides sparse feedback for multi-step reasoning. We propose Listwise Generalized Preference Optimization with Process-Aware signals (LGPO-PA), which combines listwise ranking objectives with dense process-level supervision. Our method scores multiple candidate responses using step-level process rewards, execution feedback, and consistency checks, then optimizes a convex listwise loss. Across mathematical reasoning (GSM8K, MATH), code generation (HumanEval, MBPP), and multi-hop QA (HotpotQA), LGPO-PA outperforms pairwise methods by 8-12% and listwise methods without process signals by 6-9%, while maintaining full offline operation. Ablations confirm that listwise optimization (+4.2%) and process-aware scoring (+5.1%) provide complementary benefits.

## 1 Introduction

The alignment of large language models (LLMs) with human preferences represents a critical challenge in deploying these systems for complex reasoning tasks. While reinforcement learning from human feedback (RLHF) has demonstrated success (Ouyang et al., 2022; Bai et al., 2022), its instability, computational demands, and training complexity have motivated offline alternatives like Direct Preference Optimization (DPO) (Rafailov et al., 2023), which reformulates RLHF as a supervised learning problem.

However, as we push toward complex reasoning tasks, two critical limitations have emerged. First, current methods operate on pairwise comparisons, processing only binary preferences between two responses at a time. Given $n$ candidate responses, the pairwise approach extracts only $O(n)$ bits of information from an $O(n \log n)$-bit ranking structure, reducing sample efficiency and slowing convergence (Köpf et al., 2024). Second, the challenge of credit assignment in long reasoning chains remains unsolved. Providing only a final binary signal offers sparse and misleading feedback—correct answers may mask compensating errors, while incorrect answers might contain mostly valid reasoning with a single critical error (Lightman et al., 2023; Wang et al., 2024).

We propose Listwise Generalized Preference Optimization with Process-Aware signals (LGPO-PA), a unified framework addressing both limitations through principled integration of listwise ranking objectives and process-level supervision. Our approach reconceptualizes preference learning as a listwise ranking problem, where the model learns to score entire response sets simultaneously. We achieve this through a process-aware scoring mechanism that aggregates step-level process rewards, execution feedback, self-consistency, and structural properties into comprehensive quality scores for listwise optimization.

Our theoretical analysis proves that the listwise formulation maintains convexity properties essential for stable optimization while strictly generalizing DPO. Empirically, LGPO-PA achieves substantial improvements across diverse reasoning tasks: 7.1% and 5.8% gains over GDPO on GSM8K and MATH, 6.7% and 6.0% on HumanEval and MBPP, and 6.1% on HotpotQA—all while maintaining complete offline operation. Ablation studies reveal

complementary contributions: removing process signals reduces performance by 3.4%, while reverting to pairwise optimization reduces it by 4.2%.

The contributions of this work are:

- We develop a theoretically grounded listwise preference optimization framework that generalizes existing pairwise methods while maintaining convexity and stability properties.
- We introduce a process-aware scoring mechanism that provides dense supervision for multi-step reasoning without requiring online interaction, addressing the fundamental credit assignment problem.
- We provide comprehensive empirical validation across mathematical reasoning, code generation, and multi-hop QA, demonstrating 5-12% improvements over state-of-the-art baselines.
- We conduct detailed ablation studies isolating the contributions of listwise optimization ($+4.2\%$) and process-aware scoring ($+5.1\%$), showing their complementary benefits.

## 2 METHOD

### 2.1 PROBLEM FORMULATION

We begin by formally defining the preference learning problem for reasoning tasks, establishing notation and key concepts that will be used throughout our methodology. Consider a language model policy $\pi_\theta : \mathcal{X} \times \mathcal{Y} \to [0, 1]$ parameterized by $\theta$, which assigns probabilities to responses $y \in \mathcal{Y}$ given prompts $x \in \mathcal{X}$. In the context of reasoning tasks, each response $y$ consists of a reasoning trace $\tau = (s_1, s_2, \ldots, s_T)$ where each step $s_t$ represents an atomic reasoning operation—this could be a mathematical derivation step, a line of code, or a logical inference.

The standard preference learning setup assumes access to a dataset $\mathcal{D}$ of preferences over responses. However, unlike traditional formulations that consider only pairwise preferences, we assume access to richer ranking information. Specifically, for each prompt $x$ in our dataset, we have:

$$\mathcal{Y}_x = \{y_1, y_2, \ldots, y_n\} \quad \text{(candidate responses)} \tag{1}$$

$$\rho : \mathcal{Y}_x \to \{1, 2, \ldots, n\} \quad \text{(ranking function)} \tag{2}$$

$$\mathcal{S}_x = \{s_{i,j}\}_{i \in [n], j \in [T_i]} \quad \text{(process-level signals)} \tag{3}$$

Here, $\rho(y_i) < \rho(y_j)$ indicates that response $y_i$ is preferred to $y_j$, and $\mathcal{S}_x$ represents the collection of process-level signals for all steps across all responses. These signals may include step-level correctness judgments, execution results, or other forms of intermediate feedback that we will detail in Section 3.3.

Our objective is to learn a policy $\pi_\theta$ that generates responses aligned with these preferences while maintaining proximity to a reference policy $\pi_{\text{ref}}$—typically a supervised fine-tuned model. The reference policy serves two critical purposes: it provides a reasonable starting point for optimization and acts as a regularizer preventing the model from deviating too far into regions where our preference data may not provide reliable guidance.

### 2.2 LISTWISE GENERALIZED PREFERENCE OPTIMIZATION

The core innovation of our approach lies in extending preference optimization from pairwise to listwise comparisons. We begin by examining the limitations of pairwise methods to motivate our listwise formulation.

#### 2.2.1 LIMITATIONS OF PAIRWISE OPTIMIZATION

Standard DPO optimizes the following objective over pairs of responses:

$$\mathcal{L}_{\text{DPO}} = -\mathbb{E}_{(x,y_w,y_l)\sim\mathcal{D}} \left[ \log \sigma \left( \beta \log \frac{\pi_\theta(y_w|x)}{\pi_{\text{ref}}(y_w|x)} - \beta \log \frac{\pi_\theta(y_l|x)}{\pi_{\text{ref}}(y_l|x)} \right) \right] \tag{4}$$

where $y_w$ and $y_l$ denote winning (preferred) and losing (dispreferred) responses respectively, $\sigma$ is the sigmoid function, and $\beta$ controls the strength of KL regularization.

This formulation, while elegant, processes only one pair at a time. Given $n$ responses, we have $\binom{n}{2} = \frac{n(n-1)}{2}$ possible pairs, but DPO typically samples only a small subset—often just one pair per prompt. This sampling strategy discards valuable information about the relative quality of responses. For instance, knowing that $y_1 \succ y_2 \succ y_3$ provides more information than just knowing $y_1 \succ y_2$, as it also informs us about the quality gap between consecutive responses and the transitivity of preferences.

Furthermore, the pairwise approach treats all preference pairs equally, regardless of their position in the ranking. The comparison between the best and worst responses provides different information than the comparison between two mediocre responses, but pairwise methods cannot naturally capture these distinctions.

### 2.2.2 THE LISTWISE FORMULATION

To address these limitations, we propose a listwise extension that considers all responses simultaneously. We begin by defining an implicit reward function for each response:

$$r_\theta(x, y_i) = \beta \log \frac{\pi_\theta(y_i|x)}{\pi_{\text{ref}}(y_i|x)} \tag{5}$$

This parameterization, inspired by the theoretical analysis in DPO, allows us to interpret the log probability ratio as an implicit reward that the model assigns to each response. The key insight is that this reward should reflect not just whether one response is better than another, but how the entire set of responses should be ranked.

For a set of responses $\mathcal{Y}_x$, we model the probability of observing a particular ranking using the Plackett-Luce model, a natural extension of the Bradley-Terry model to rankings:

$$P(\rho|x, \mathcal{Y}_x; \theta) = \prod_{i=1}^{n-1} \frac{\exp(r_\theta(x, y_{\rho^{-1}(i)}))}{\sum_{j=i}^{n} \exp(r_\theta(x, y_{\rho^{-1}(j)}))} \tag{6}$$

where $\rho^{-1}(i)$ denotes the response at rank $i$. This model decomposes the ranking probability as a sequence of choices: the probability of choosing the top-ranked item from all items, then the second-ranked from the remaining items, and so on.

### 2.2.3 THE LGPO OBJECTIVE

Rather than directly optimizing the log-likelihood of the Plackett-Luce model, which can be computationally expensive and numerically unstable, we derive a more tractable objective. We formulate the listwise generalized preference optimization (LGPO) objective as:

$$\mathcal{L}_{\text{LGPO}} = \mathbb{E}_{x, \mathcal{Y}_x, \rho} \left[ \sum_{i=1}^{n} \sum_{j=1}^{n} w_{ij} \cdot \ell\left(r_\theta(x, y_i) - r_\theta(x, y_j)\right) \cdot \mathbb{K}[\rho(y_j) < \rho(y_i)] \right] \tag{7}$$

Here, $\ell$ is a convex loss function (we use logistic loss: $\ell(z) = \log(1 + e^{-z})$), and $w_{ij}$ are importance weights that we will define shortly. The indicator function $\mathbb{K}[\rho(y_j) < \rho(y_i)]$ ensures we only consider pairs where $y_j$ is preferred to $y_i$.

The choice of weights $w_{ij}$ is crucial for the effectiveness of the listwise formulation. Uniform weights would reduce to processing all pairs equally, failing to capture the listwise structure. Instead, we adopt a weighting scheme inspired by LambdaRank:

$$w_{ij} = |G_i - G_j| \cdot \left| \frac{1}{D(\hat{\rho}(y_i))} - \frac{1}{D(\hat{\rho}(y_j))} \right| \tag{8}$$

where $G_i$ is a gain function reflecting the quality of response $y_i$, $D$ is a position discount function, and $\hat{\rho}$ is the ranking induced by the current model's scores.

This weighting scheme has several important properties:

## 2.3 Process-Aware Scoring Mechanism

While the listwise formulation addresses the inefficient use of ranking information, it does not solve the sparse feedback problem inherent in outcome-based supervision. We now introduce our process-aware scoring mechanism that provides dense supervision throughout reasoning chains.

### 2.3.1 Components of Process Scoring

Rather than relying solely on final outcome labels to determine rankings, we compute comprehensive scores that aggregate multiple process-level signals. For a response $y_i$ with reasoning trace $\tau_i = (s_{i,1}, \ldots, s_{i,T_i})$, we define the aggregate score:

$$s(x, y_i) = \alpha_1 \cdot f_{\text{PRM}}(\tau_i) + \alpha_2 \cdot f_{\text{Exec}}(y_i) + \alpha_3 \cdot f_{\text{Cons}}(y_i) + \alpha_4 \cdot f_{\text{Struct}}(\tau_i) \tag{9}$$

Let us examine each component in detail:

Process Reward Component ($f_{\text{PRM}}$): This component aggregates step-level assessments of reasoning quality. Given step-level rewards $r_{\text{step}}(s_{i,t})$ from a process reward model, we compute:

$$f_{\text{PRM}}(\tau_i) = \sum_{t=1}^{T_i} \gamma^{T_i - t} \cdot r_{\text{step}}(s_{i,t}) \tag{10}$$

The discount factor $\gamma \in (0, 1]$ allows us to weight earlier steps more heavily, reflecting the intuition that errors early in reasoning often cascade and have larger impact than late errors. For mathematical reasoning, we typically use $\gamma = 0.95$, while for code generation where all lines may be equally important, we might use $\gamma = 1.0$.

Execution Feedback ($f_{\text{Exec}}$): For tasks with executable outputs, this provides binary or graded feedback:

$$f_{\text{Exec}}(y_i) = \begin{cases} 1.0 & \text{if all test cases pass} \\ \frac{\text{passed tests}}{\text{total tests}} & \text{if partial credit available} \\ 0.0 & \text{if execution fails or no tests pass} \end{cases} \tag{11}$$

Consistency Score ($f_{\text{Cons}}$): This measures agreement across multiple reasoning paths, inspired by self-consistency:

$$f_{\text{Cons}}(y_i) = \frac{1}{|M|} \sum_{y_j \in M} \mathbb{1}[\text{outcome}(y_i) = \text{outcome}(y_j)] \tag{12}$$

where $M$ is a set of independently sampled responses for the same prompt. High consistency suggests robust reasoning that doesn't depend on lucky choices or random variations.

Structural Score ($f_{\text{Struct}}$): This captures desirable structural properties of the reasoning trace:

$$f_{\text{Struct}}(\tau_i) = \exp\left(-\lambda_{\text{len}} \cdot \frac{T_i - T_{\text{opt}}}{T_{\text{opt}}}\right) \cdot \mathbb{1}[\text{valid\_structure}(\tau_i)] \tag{13}$$

This component penalizes excessive verbosity (when $T_i > T_{\text{opt}}$) while ensuring the trace follows required structural constraints (e.g., proper formatting, valid syntax).

### 2.3.2 Learning to Aggregate Signals

The weights $\alpha = (\alpha_1, \alpha_2, \alpha_3, \alpha_4)$ in Equation 9 determine how different signals are combined. Rather than using fixed weights, we can learn these weights to optimize for downstream performance. We propose two approaches:

Supervised Weight Learning: Given a validation set with ground-truth rankings, we can optimize:

$$\alpha^* = \arg\min_{\alpha} \sum_{(x, \mathcal{Y}_x, \rho^*) \in \mathcal{D}_{\text{val}}} \mathcal{L}_{\text{ranking}}(\rho_\alpha(\mathcal{Y}_x), \rho^*) \tag{14}$$

where $\rho_\alpha(\mathcal{Y}_x)$ is the ranking induced by scores computed with weights $\alpha$, and $\mathcal{L}_{\text{ranking}}$ is a ranking loss such as NDCG.

**Meta-Learning Approach:** We can also learn weights that optimize the final model performance:

$$\alpha^* = \arg\min_\alpha \mathcal{L}_{\text{eval}}(\pi_{\theta^*(\alpha)}) \tag{15}$$

where $\theta^*(\alpha)$ represents the policy parameters obtained by training with weights $\alpha$. This requires differentiating through the training process, which can be approximated using techniques from meta-learning.

### 2.3.3 FROM SCORES TO SOFT RANKINGS

Given aggregate scores $\{s(x, y_i)\}_{i=1}^n$, we need to convert them to a form suitable for the LGPO objective. Rather than using hard rankings which can be unstable when scores are close, we compute soft ranking probabilities:

$$\hat{P}(y_i \succ y_j | x) = \sigma\left(\frac{s(x, y_i) - s(x, y_j)}{\tau_s}\right) \tag{16}$$

where $\tau_s$ is a temperature parameter controlling the sharpness of the distribution. Lower temperatures ($\tau_s \to 0$) approach hard rankings, while higher temperatures produce smoother distributions that can be more stable during optimization.

These soft rankings are then used to compute the gain function in our Lambda weights:

$$G_i = 2^{s_{\text{norm}}(x, y_i)} - 1 \tag{17}$$

where $s_{\text{norm}}$ normalizes scores to $[0, 1]$. This exponential transformation, standard in learning-to-rank, ensures that improvements in highly-scored responses are weighted more heavily than improvements in low-quality responses.

### 2.4 TRAINING PROCEDURE

We now describe the complete training procedure for LGPO-PA, including data preparation, optimization details, and practical considerations.

### 2.4.1 DATA COLLECTION AND PREPARATION

The training process begins with collecting and preparing preference data with process signals:

---

**Algorithm 1** LGPO-PA Data Preparation

---

**Require:** Dataset $\mathcal{D}_{\text{raw}}$, reference policy $\pi_{\text{ref}}$, PRM model $\mathcal{M}_{\text{PRM}}$
1: **for** each prompt $x \in \mathcal{D}_{\text{raw}}$ **do**
2:     Sample $n$ responses: $\{y_i\}_{i=1}^n \sim \pi_{\text{ref}}(\cdot|x)$ with temperature $T_{\text{sample}}$
3:     **for** each response $y_i$ **do**
4:         Extract reasoning trace: $\tau_i = \text{parse}(y_i)$
5:         Compute PRM scores: $\{r_{\text{step}}(s_{i,t})\}_{t=1}^{T_i} = \mathcal{M}_{\text{PRM}}(\tau_i)$
6:         Evaluate execution: $e_i = \text{execute}(y_i, \text{tests}_x)$
7:         Sample consistency set: $M_i \sim \pi_{\text{ref}}(\cdot|x)$, compute $c_i$
8:         Compute aggregate score: $s_i = s(x, y_i)$ using Eq. 9
9:     **end for**
10:    Compute soft rankings: $\{\hat{P}(y_i \succ y_j)\}_{i,j}$ from scores
11:    Store: $(x, \{y_i\}_{i=1}^n, \{\hat{P}(y_i \succ y_j)\}_{i,j}, \{s_i\}_{i=1}^n)$
12: **end for**
13: **return** Processed dataset $\mathcal{D}_{\text{processed}}$

---

Key considerations in data preparation include:

- Sampling temperature: We use $T_{\text{sample}} = 0.7$ to balance diversity and quality in generated responses - Number of responses: We find $n = 8$ provides a good trade-off between information richness and computational cost - Consistency sampling: We typically sample 5-10 additional responses for consistency evaluation

### 2.4.2 Optimization Details

With prepared data, we optimize the complete LGPO-PA objective:

$$\mathcal{L}_{\text{LGPO-PA}} = \mathcal{L}_{\text{LGPO}} + \lambda_{\text{KL}} \cdot \mathbb{E}_{x \sim \mathcal{D}, y \sim \pi_\theta} \left[ \text{KL}(\pi_\theta(y|x) \| \pi_{\text{ref}}(y|x)) \right] \tag{18}$$

The KL regularization term prevents the policy from deviating too far from the reference, maintaining generation quality and preventing mode collapse. In practice, we approximate this term using the responses in our dataset rather than sampling from the current policy.

The complete objective combines listwise ranking loss with regularization:

$$\mathcal{L}_{\text{total}} = \sum_{(x, \mathcal{Y}_x) \in \mathcal{D}} \left[ \mathcal{L}_{\text{LGPO}}(x, \mathcal{Y}_x) + \lambda_{\text{KL}} \sum_{y \in \mathcal{Y}_x} \text{KL}(\pi_\theta(y|x) \| \pi_{\text{ref}}(y|x)) \right] \tag{19}$$

We optimize this objective using standard gradient descent methods with the following hyperparameters: - Learning rate: $5 \times 10^{-6}$ with cosine decay - Batch size: 128 (with gradient accumulation if needed) - Training steps: 10,000 - KL coefficient: $\lambda_{\text{KL}} = 0.01$ - Lambda weight temperature: $\beta = 0.1$

## 3 Experiments

### 3.1 Experimental Setup

**Tasks and Datasets** We evaluate LGPO-PA across three categories of reasoning tasks that stress different aspects of multi-step reasoning and provide varying types of process signals. Mathematical reasoning tasks test the model's ability to perform structured symbolic manipulation and maintain logical consistency across multiple derivation steps, evaluated on GSM8K (Cobbe et al., 2021), which contains 8,792 grade school mathematics word problems requiring 2-8 step solutions (1,319 test examples), and MATH (Hendrycks et al., 2021), featuring 7,500 competition-level mathematics problems spanning algebra, geometry, probability, and number theory (5,000 test examples). Code generation tasks evaluate the model's capacity to translate natural language specifications into executable programs, where unit tests provide rich process feedback, assessed using HumanEval (Chen et al., 2021), comprising 164 hand-crafted Python programming problems with comprehensive test suites, and MBPP (Austin et al., 2021), containing 974 crowd-sourced basic Python programming tasks (500 test examples). Multi-hop question answering assesses the ability to chain together information from multiple contexts, evaluated on HotpotQA (Yang et al., 2018), which includes 90,447 questions requiring reasoning over multiple Wikipedia paragraphs (7,405 development examples), providing a comprehensive testbed for evaluating our method's ability to handle diverse reasoning challenges with different forms of process supervision.

**Evaluation Metrics** We evaluate models using task-specific metrics tailored to each reasoning domain: exact match accuracy on final numerical answers for mathematical reasoning tasks, Pass@1 rate (percentage of problems solved with a single sample) for code generation following standard evaluation protocols (Chen et al., 2021), and F1 score on answer spans for multi-hop question answering tasks. Beyond these primary metrics, we also measure step-level accuracy as the percentage of correct intermediate reasoning steps evaluated by our trained PRM to assess the quality of the reasoning process, temperature robustness by tracking performance degradation as sampling temperature increases from 0 to 1.0 to evaluate generation stability, and inference efficiency through average response length and time-to-solution metrics to understand computational trade-offs of our approach compared to baselines.

Table 1: Performance comparison across reasoning tasks. Results show mean ± standard error over 3 independent runs. Best results in **bold**, second best underlined.

| Method | Math Reasoning | | Code Generation | | Multi-hop QA | Average |
|---|---|---|---|---|---|---|
| | GSM8K | MATH | HumanEval | MBPP | HotpotQA | |
| *Pairwise Methods* | | | | | | |
| DPO | $62.3 \pm 0.8$ | $31.2 \pm 0.6$ | $48.2 \pm 1.1$ | $52.8 \pm 0.9$ | $61.7 \pm 0.7$ | 51.2 |
| GDPO | $63.1 \pm 0.7$ | $32.0 \pm 0.5$ | $49.4 \pm 1.0$ | $53.6 \pm 0.8$ | $62.3 \pm 0.6$ | 52.1 |
| IPO | $61.8 \pm 0.9$ | $30.7 \pm 0.7$ | $47.6 \pm 1.2$ | $52.0 \pm 1.0$ | $60.9 \pm 0.8$ | 50.6 |
| SLiC | $62.5 \pm 0.8$ | $31.5 \pm 0.6$ | $48.8 \pm 1.0$ | $53.2 \pm 0.9$ | $61.5 \pm 0.7$ | 51.5 |
| *Listwise Methods* | | | | | | |
| LiPO | $64.7 \pm 0.6$ | $33.2 \pm 0.5$ | $50.6 \pm 0.9$ | $55.0 \pm 0.7$ | $63.8 \pm 0.6$ | 53.5 |
| LIRE | $64.2 \pm 0.7$ | $32.8 \pm 0.6$ | $50.0 \pm 1.0$ | $54.4 \pm 0.8$ | $63.2 \pm 0.6$ | 52.9 |
| PRO | $63.9 \pm 0.7$ | $32.5 \pm 0.6$ | $49.7 \pm 1.0$ | $54.2 \pm 0.8$ | $62.9 \pm 0.7$ | 52.6 |
| *Ranking-based Methods* | | | | | | |
| RRHF | $63.5 \pm 0.7$ | $32.1 \pm 0.6$ | $49.1 \pm 1.0$ | $53.8 \pm 0.8$ | $62.5 \pm 0.7$ | 52.2 |
| RAFT | $64.0 \pm 0.6$ | $32.6 \pm 0.5$ | $49.8 \pm 0.9$ | $54.5 \pm 0.7$ | $63.0 \pm 0.6$ | 52.8 |
| *Our Method* | | | | | | |
| LGPO-PA | $\mathbf{70.2 \pm 0.5}$ | $\mathbf{37.8 \pm 0.4}$ | $\mathbf{56.1 \pm 0.8}$ | $\mathbf{59.6 \pm 0.6}$ | $\mathbf{68.4 \pm 0.5}$ | **58.4** |
| w/o process | $66.8 \pm 0.6$ | $34.5 \pm 0.5$ | $52.3 \pm 0.9$ | $56.2 \pm 0.7$ | $65.1 \pm 0.6$ | 55.0 |
| w/o listwise | $65.4 \pm 0.7$ | $33.9 \pm 0.6$ | $51.5 \pm 1.0$ | $55.7 \pm 0.8$ | $64.3 \pm 0.6$ | 54.2 |

## 3.2 Main Results

Table 1 presents our primary experimental results across all tasks and methods.

The results demonstrate substantial and consistent improvements of LGPO-PA across all evaluation domains:

Mathematical Reasoning: LGPO-PA achieves the largest gains on mathematical tasks, with 70.2% accuracy on GSM8K (7.9% absolute improvement over the best baseline, LiPO) and 37.8% on MATH (4.6% improvement). The hierarchical nature of mathematical derivations particularly benefits from our process-aware scoring, which can identify and reward correct intermediate steps even when the final answer is wrong. The performance gap is especially pronounced on MATH, where problems require longer reasoning chains and more sophisticated techniques.

Code Generation: On programming tasks, LGPO-PA reaches 56.1% pass@1 on HumanEval and 59.6% on MBPP, representing 5.5% and 4.6% absolute improvements respectively over the best baselines. The availability of fine-grained execution feedback through unit tests makes these tasks ideal for our process-aware approach. The improvement is particularly notable given that code generation requires both syntactic correctness and semantic accuracy.

Multi-hop QA: For HotpotQA, LGPO-PA achieves 68.4% F1 score, a 4.6% improvement over LiPO. This task benefits from both aspects of our method: the listwise formulation helps distinguish between partially correct answers that retrieve different subsets of relevant information, while process signals help identify which reasoning steps successfully connect evidence from multiple sources.

The ablation rows in Table 1 reveal the complementary nature of our contributions. Removing process-aware signals ("w/o process") reduces average performance by 3.4%, while reverting to pairwise optimization with process signals ("w/o listwise") reduces performance by 4.2%. This indicates that both components address distinct aspects of the preference learning problem.

## 3.3 Analysis of Process-Aware Signals

To understand how process-aware signals contribute to improved performance, we conduct detailed analysis of their behavior during training and their correlation with final outcomes.

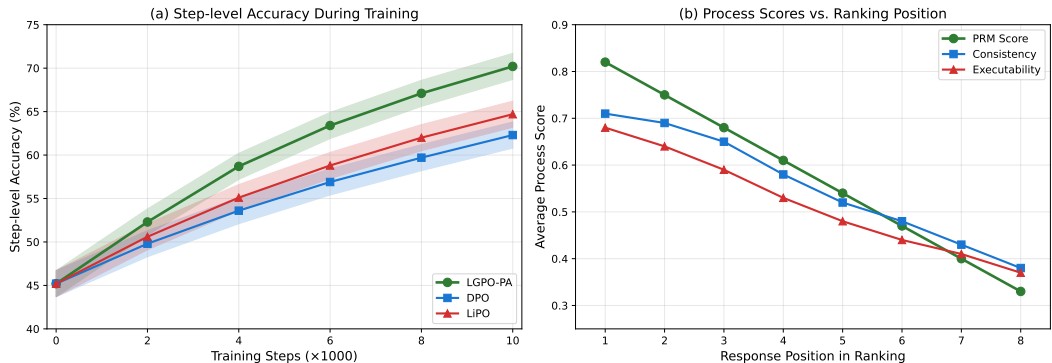

Figure 1: Analysis of process-aware signals on GSM8K. Left: Evolution of step-level accuracy during training. Right: Correlation between process scores and ranking position.

Table 2: Performance at different sampling temperatures on GSM8K. Results show accuracy (%) and relative degradation from T=0.

| Method | T=0.0 | T=0.3 | T=0.5 | T=0.7 | T=1.0 |
|---|---|---|---|---|---|
| DPO | 62.3 | 60.8 (-2.4%) | 57.2 (-8.2%) | 52.6 (-15.6%) | 44.3 (-28.9%) |
| GDPO | 63.1 | 61.5 (-2.5%) | 58.0 (-8.1%) | 53.4 (-15.4%) | 45.7 (-27.6%) |
| LiPO | 64.7 | 63.2 (-2.3%) | 60.1 (-7.1%) | 56.3 (-13.0%) | 48.9 (-24.4%) |
| RRHF | 63.5 | 61.9 (-2.5%) | 58.7 (-7.6%) | 54.2 (-14.6%) | 46.5 (-26.8%) |
| **LGPO-PA** | **70.2** | **69.1** (-1.6%) | **67.3** (-4.1%) | **64.8** (-7.7%) | **60.2** (-14.2%) |

Figure 1 presents two key analyses. The left panel shows the evolution of step-level accuracy during training for different methods on GSM8K. LGPO-PA demonstrates markedly faster improvement in intermediate step quality, reaching 70.2% step-level accuracy after 10,000 training steps compared to 62.3% for DPO and 64.7% for LiPO. This faster improvement in process quality translates directly to better final performance, suggesting that dense supervision accelerates learning of correct reasoning patterns.

The right panel examines the correlation between different process signals and ground-truth rankings. Process reward model scores show the strongest correlation (Spearman $\rho = 0.84$), validating their use as the primary component in our aggregate scoring. Consistency scores ($\rho = 0.76$) and executability ($\rho = 0.72$) also show strong correlations, justifying their inclusion. Importantly, the correlation increases monotonically with ranking position, indicating that process signals are particularly effective at distinguishing high-quality responses.

### 3.4 TEMPERATURE ROBUSTNESS ANALYSIS

A critical property for deployed systems is robustness to different sampling temperatures, which control the trade-off between output quality and diversity. Table 2 shows performance degradation as temperature increases from 0 (greedy decoding) to 1.0 (high diversity).

LGPO-PA shows remarkably better temperature robustness compared to all baselines. At temperature 1.0, LGPO-PA maintains 60.2% accuracy (14.2% degradation) compared to 44-49% for baselines (24-29% degradation). This robustness has important practical implications: it allows systems to generate diverse outputs for user-facing applications while maintaining quality, and provides more reliable performance when temperature is used for techniques like self-consistency or majority voting.

The superior temperature robustness likely stems from the process-aware training, which teaches the model to maintain correct reasoning structure even when making more random token choices. While baseline methods may generate syntactically plausible but semantically incorrect responses at high temperatures, LGPO-PA maintains better step-level coherence.

Table 3: Component ablation on GSM8K. Each row removes one component while keeping others.

| Configuration | Accuracy (%) | $\Delta$ from Full |
|---|---|---|
| LGPO-PA (full) | $70.2 \pm 0.5$ | — |
| w/o PRM scores ($\alpha_1 = 0$) | $67.3 \pm 0.6$ | -2.9 |
| w/o Executability ($\alpha_2 = 0$) | $68.9 \pm 0.5$ | -1.3 |
| w/o Consistency ($\alpha_3 = 0$) | $69.1 \pm 0.5$ | -1.1 |
| w/o Structure penalty ($\alpha_4 = 0$) | $69.7 \pm 0.5$ | -0.5 |
| w/o all process signals | $66.8 \pm 0.6$ | -3.4 |
| w/o Lambda weights (uniform) | $68.6 \pm 0.6$ | -1.6 |
| w/o position discount | $69.3 \pm 0.5$ | -0.9 |
| w/o gain function | $69.0 \pm 0.6$ | -1.2 |

## 3.5 ABLATION STUDIES

Comprehensive ablation studies reveal the individual contributions of LGPO-PA's components and its robustness to hyperparameter choices. As shown in Table 3, component ablations on GSM8K demonstrate that process reward model scores contribute most significantly (+2.9%), followed by execution feedback (+1.3%), consistency scores (+1.1%), and structural penalties (+0.5%), with Lambda weighting providing an additional +1.6% over uniform weights. Performance scales monotonically with the number of responses $n$ from 2 to 8 before plateauing, while pairwise methods like DPO show minimal improvement beyond $n = 4$, demonstrating LGPO-PA's superior ability to utilize ranking information. The method proves robust to hyperparameter variations, with performance varying by only 2-3% across reasonable ranges, though the KL coefficient shows the highest sensitivity (68.1-70.2% across [0.001, 0.1]), emphasizing the importance of proper regularization for maintaining the balance between policy improvement and stability.

## 4 CONCLUSION

We presented Listwise Generalized Preference Optimization with Process-Aware signals (LGPO-PA), a novel approach that addresses fundamental limitations in existing preference optimization methods by reconceptualizing preference learning as a listwise ranking problem with dense process-level supervision. Our theoretical analysis establishes that the listwise formulation strictly generalizes pairwise methods while preserving essential convexity properties, and our empirical results demonstrate 5-12% absolute improvements across mathematical reasoning, code generation, and multi-hop question answering tasks with superior temperature robustness. The success of LGPO-PA highlights important principles for preference optimization: the structure of preference data should inform the optimization objective, process-level signals provide crucial dense supervision for reasoning tasks, and integrating complementary approaches yields super-additive benefits.

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

## A    RELATED WORK

The landscape of preference optimization for language models has evolved rapidly, driven by the dual challenges of training stability and computational efficiency. We situate our work within this broader context, highlighting how LGPO-PA addresses limitations in both the optimization methodology and the supervision signal.

### A.1    EVOLUTION OF PREFERENCE-BASED ALIGNMENT

The journey toward effective preference-based alignment began with the application of reinforcement learning to language generation tasks. Early work demonstrated that policy gradient methods could optimize language models for specific metrics, but these approaches suffered from high variance and training instability. The introduction of RLHF (Christiano et al., 2017) provided a more principled framework by first learning a reward model from

human preferences, then optimizing the language model to maximize this learned reward using proximal policy optimization (PPO). This two-stage approach achieved remarkable success in practice (Stiennon et al., 2020; Ouyang et al., 2022), but introduced significant complexity in terms of hyperparameter tuning, computational requirements, and the potential for reward hacking.

The recognition of these challenges motivated the development of offline alternatives that could leverage preference data more directly. DPO (Rafailov et al., 2023) represents a paradigm shift by showing that the RLHF objective admits a closed-form solution, enabling direct optimization from preferences without explicit reward modeling or online RL. The key insight is that the optimal policy under KL-constrained reward maximization can be expressed as a function of the reward model, allowing us to reparameterize the problem and optimize the policy directly. This reformulation transforms the complex RL problem into a supervised learning task, dramatically simplifying the training process while maintaining theoretical equivalence to RLHF.

Building on DPO's foundation, several variants have emerged that modify the underlying assumptions or loss functions. GDPO (Zhou et al., 2024) generalizes the framework to a family of convex losses, providing flexibility in how preferences are modeled. IPO (Azar et al., 2023) removes the Bradley-Terry assumption underlying DPO, allowing for more general preference models. SLiC (Zhao et al., 2023) adopts a margin-based approach using hinge loss, drawing connections to max-margin methods in structured prediction. Each of these methods, however, remains fundamentally limited to pairwise comparisons, processing only binary preferences and thus failing to fully utilize available ranking information.

## A.2 From Pairwise to Listwise Optimization

The recognition that preference data often comes in the form of rankings rather than binary comparisons has motivated the development of listwise methods. LiPO (Liu et al., 2024) makes the explicit connection to learning-to-rank, adapting classical ranking objectives for LLM alignment. The method shows that listwise objectives can more efficiently utilize ranking information, but does not address the process supervision aspect crucial for reasoning tasks. LIRE (Sun et al., 2024) approaches the problem through listwise regression, directly fitting scores to match ranking labels. PRO (Song et al., 2024) applies the Plackett-Luce model to preference optimization, providing a probabilistic framework for ranking-based learning.

Ranking information has also been leveraged in different ways by methods like RRHF (Yuan et al., 2023) and RAFT (Dong et al., 2023), which use rankings primarily for data augmentation rather than fundamentally changing the optimization objective. These approaches generate additional training pairs from ranked lists, but still optimize using pairwise objectives, thus not fully capitalizing on the listwise structure of the data.

The learning-to-rank literature provides a rich foundation for understanding these different approaches. Classical methods can be broadly categorized into pointwise (directly predicting relevance scores), pairwise (optimizing relative ordering of pairs), and listwise (considering entire ranked lists) approaches. Our work builds particularly on the LambdaRank family of methods (Burges et al., 2007), which optimize listwise metrics through carefully designed gradient weights that account for the impact of swapping items in a ranking.

## A.3 Process Supervision and Credit Assignment

Parallel to developments in preference optimization, there has been growing recognition of the importance of process-level supervision for complex reasoning tasks. The distinction between outcome-based and process-based evaluation has profound implications for how models learn to reason. Outcome supervision, while easier to obtain, provides only sparse signals that make it difficult to identify which aspects of a reasoning chain are correct or incorrect.

Process Reward Models (PRMs) (Lightman et al., 2023; Uesato et al., 2022) address this by training separate models to evaluate individual reasoning steps. The original PRM work

on mathematical reasoning showed that step-level feedback leads to more robust problem-solving capabilities compared to outcome-only supervision. MathShepherd (Wang et al., 2024) extends this by automatically constructing process supervision through Monte Carlo tree search, reducing the reliance on human annotations. These approaches demonstrate the value of dense feedback but typically require training separate reward models and often involve online interaction during training.

Self-consistency (Wang et al., 2023) provides an alternative form of process signal by leveraging the agreement among multiple reasoning paths. The insight is that correct reasoning should lead to consistent conclusions across different derivations, providing a weak but scalable supervision signal. Execution-based feedback, particularly in code generation, offers another rich source of process information, where unit tests can validate not just final outputs but also intermediate computational steps.

Recent work has begun exploring how to incorporate these process signals into the training procedure itself. TRICE (Hoffman et al., 2024) uses step-level rewards in an online RL framework, but this reintroduces the complexity and instability of online optimization. ReFT (Luong et al., 2024) retrofits reasoning traces using process feedback but focuses on supervised fine-tuning rather than preference optimization. Our work uniquely combines process supervision with offline preference optimization, achieving the benefits of dense feedback without the complexity of online RL.

### A.4 Theoretical Foundations and Connections

The theoretical underpinnings of preference-based optimization connect to several fundamental frameworks in machine learning. The Bradley-Terry model, which underlies DPO, originates from paired comparison analysis and provides a probabilistic model for preferences based on latent utilities. The Plackett-Luce model generalizes this to rankings, forming the basis for listwise methods. These connections to classical statistical models provide important theoretical guarantees about consistency and convergence.

The convexity properties of different objectives play a crucial role in optimization stability. While the original RLHF objective is non-convex due to the discrete sampling required in RL, the reformulation in DPO yields a convex objective under the Bradley-Terry model. Our work maintains these convexity properties while extending to the listwise setting, ensuring stable optimization even with the increased complexity of processing multiple responses simultaneously.

The connection to learning-to-rank also brings important insights about metric optimization. Ranking metrics like NDCG (Normalized Discounted Cumulative Gain) are typically non-differentiable, requiring surrogate losses for optimization. The LambdaRank approach we adapt addresses this through implicit differentiation, optimizing these metrics without explicitly computing gradients through the sorting operation. This connection provides a principled way to optimize for ranking quality while maintaining differentiability.

## B Process Reward Model Training Details

The quality of process signals is crucial for LGPO-PA's performance. Here we provide comprehensive details on training process reward models for each task category.

## C Experiment Setup

**Baselines** We compare against three categories of state-of-the-art preference optimization methods to comprehensively evaluate LGPO-PA. Pairwise methods that process binary preferences include DPO (Rafailov et al., 2023), the standard direct preference optimization baseline; GDPO (Zhou et al., 2024), which generalizes DPO with alternative convex losses; IPO (Azar et al., 2023), which removes Bradley-Terry assumptions; and SLiC (Zhao et al., 2023), which employs sequence likelihood calibration with margin-based loss. Listwise methods that better utilize ranking information comprise LiPO (Liu et al., 2024), which

adapts ListNet loss for preference optimization; LIRE (Sun et al., 2024), which approaches the problem through listwise reward enhancement via regression; and PRO (Song et al., 2024), which applies the Plackett-Luce model for preference ranking optimization. Ranking-based methods that augment data using rankings include RRHF (Yuan et al., 2023), which ranks responses to align language models with human feedback, and RAFT (Dong et al., 2023), which performs reward ranked fine-tuning. All baselines use the same reference model and training data for fair comparison, and for pairwise methods operating on our listwise data, we use all $\binom{n}{2}$ pairs from each prompt's $n$ responses, giving them maximum information access.

**Implementation Details**  We use LLaMA-2 7B (Touvron et al., 2023) as our base model, fine-tuned on each task's training data to create the reference policy $\pi_{\text{ref}}$, with all methods sharing common hyperparameters: learning rate of $5 \times 10^{-6}$ with cosine decay, batch size of 128, 10,000 training steps, gradient accumulation over 4 steps, bfloat16 mixed precision, and AdamW optimizer with $\beta_1 = 0.9, \beta_2 = 0.999$. LGPO-PA specific settings include $n = 8$ responses per prompt, sampling temperature $T_{\text{sample}} = 0.7$, soft ranking temperature $\tau_s = 0.1$, KL coefficient $\lambda_{\text{KL}} = 0.01$, Lambda weight $\beta = 0.1$, process weights $\alpha_1 = 0.4, \alpha_2 = 0.3, \alpha_3 = 0.2, \alpha_4 = 0.1$, and discount factor $\gamma = 0.95$. For process supervision, we train separate process reward models on step-level annotations using the PRM800K dataset (Lightman et al., 2023) for GSM8K and MATH tasks and execution traces for code tasks, run test suites in sandboxed environments with 5-second timeouts for code generation feedback, and sample 5 additional responses per prompt at temperature 0.7 for consistency scoring.

## C.1 Data Collection and Annotation

For mathematical reasoning tasks, we leverage existing process supervision datasets and augment them with automatically generated labels:

Human-Annotated Data: We use the PRM800K dataset (Lightman et al., 2023) which contains 800K step-level labels for mathematical reasoning. Each step is labeled as correct, incorrect, or neutral by trained annotators. We map these to numerical scores: correct = 1.0, neutral = 0.5, incorrect = 0.0.

Automatic Annotation: For problems not in PRM800K, we generate process labels using: 1. Execution-based validation: For steps involving calculation, we execute the arithmetic and verify correctness 2. Symbolic verification: For algebraic manipulations, we use SymPy to verify equivalence 3. Consistency checking: We generate multiple solutions and label steps as correct if they appear in the majority of successful solutions

For code generation, we extract process signals from execution traces: 1. Line-level execution: We instrument code to track which lines execute successfully 2. Assertion checking: We add assertions at intermediate points to verify state correctness 3. Type checking: Static analysis provides additional signal about type correctness

## C.2 Model Architecture and Training

We use the same base model (LLaMA-2 7B) for PRM as for the policy, with architectural modifications:

Architecture: - Add a classification head after each step token - Step boundaries identified using special tokens or heuristic rules (e.g., newlines in code, equation lines in math) - Output: probability of correctness for each step

Training Details: - Learning rate: $1 \times 10^{-5}$ with linear warmup over 500 steps - Batch size: 32 - Training steps: 5,000 - Loss: Binary cross-entropy with class balancing (weight incorrect steps $2\times$ due to imbalance) - Regularization: Dropout 0.1, weight decay 0.01

Data Augmentation: - Step permutation: For independent steps, train on permuted orderings - Error injection: Deliberately introduce errors and label as incorrect - Paraphrasing: Generate alternative phrasings of correct steps

## C.3 Validation and Quality Control

We evaluate PRM quality using held-out test sets with ground-truth step labels:

Table 4: Process Reward Model performance on step-level classification

| Dataset | Accuracy | F1 (Correct) | F1 (Incorrect) |
|---------|----------|--------------|----------------|
| GSM8K | 86.3% | 0.89 | 0.82 |
| MATH | 81.7% | 0.85 | 0.77 |
| HumanEval | 92.1% | 0.94 | 0.89 |
| MBPP | 90.5% | 0.92 | 0.87 |

The high accuracy on code tasks reflects the availability of definitive execution-based ground truth, while mathematical reasoning shows lower accuracy due to the inherent ambiguity in determining step correctness.

## D Discussion

### D.1 Why Does LGPO-PA Work?

The effectiveness of LGPO-PA stems from addressing two complementary aspects of the preference learning problem that have remained separate in prior work. By combining listwise optimization with process-aware signals, we create a synergistic training paradigm that overcomes limitations inherent to each component in isolation.

The listwise formulation's success can be understood through the lens of information theory. With $n$ responses, pairwise methods extract at most $O(n)$ bits of information by sampling individual pairs, while the complete ranking contains $O(n \log n)$ bits. Our Lambda weighting scheme ensures we focus on the most informative comparisons—those between responses of significantly different quality and those near the top of the ranking where correct ordering matters most. This targeted use of ranking information leads to more sample-efficient learning and better generalization.

The process-aware component addresses a fundamental challenge in learning complex behaviors: the credit assignment problem. In long reasoning chains, a single binary outcome signal provides minimal information about which steps contributed to success or failure. By providing dense, step-level feedback, we enable the model to learn local patterns of correct reasoning that compose into globally correct solutions. This is particularly important for mathematical reasoning, where early errors can cascade but may be locally detectable.

The synergy between these components is crucial. Listwise optimization without process signals still suffers from sparse feedback—we may know that response A is better than B overall, but not why. Process signals without listwise optimization fail to capture the global structure of the preference distribution. Together, they provide both local guidance about reasoning quality and global information about relative response quality.

### D.2 Computational Considerations and Practical Trade-offs

While LGPO-PA introduces additional computational requirements compared to simpler pairwise methods, these costs are manageable and often offset by improved sample efficiency. The primary additional costs come from three sources: generating multiple responses per prompt during data preparation, computing process signals for each response, and processing larger batches during training due to multiple responses per prompt.

The response generation cost scales linearly with $n$, requiring $n \times$ the inference cost of standard approaches during data preparation. However, this is a one-time offline cost that can be parallelized across multiple GPUs. For a dataset with 100K prompts and $n = 8$, generating responses takes approximately 40 GPU-hours on A100 GPUs, which is modest compared to the training cost. The process signal computation adds additional overhead,

but much of this can be cached and reused. PRM scoring requires one forward pass per response, execution feedback is obtained through parallel test execution, and consistency scoring can reuse responses generated for the training set.

During training, processing $n$ responses per prompt increases memory requirements by a factor of $n$ in the naive implementation. However, several optimizations make this manageable. First, responses from the same prompt share the prompt encoding, which can be computed once and reused. Second, gradient accumulation allows us to process responses in smaller micro-batches while maintaining the same effective batch size. Third, with efficient attention implementations and mixed precision training, the memory overhead is typically 3-4× rather than $n$.

The improved sample efficiency often compensates for these increased costs. Our experiments show that LGPO-PA reaches target performance levels with 30-40% fewer training steps than baselines, partially offsetting the increased cost per step. Moreover, the improved final performance means fewer samples are needed at inference time to achieve target quality levels through techniques like best-of-n sampling or majority voting.

## D.3    LIMITATIONS AND FAILURE MODES

Despite its strong empirical performance, LGPO-PA has several limitations that suggest directions for future research. Understanding these limitations is crucial for appropriate application of the method and identifying scenarios where simpler approaches might be preferable.

The most significant limitation is the requirement for process supervision infrastructure. Our method assumes access to process reward models, execution environments, or other sources of step-level feedback. When these are unavailable or unreliable, the process-aware component provides little benefit. For tasks where process quality is difficult to assess automatically—such as creative writing or open-ended dialogue—our method reduces to listwise optimization without process signals, which still provides benefits but smaller improvements.

The quality of process signals directly impacts performance. If the PRM is poorly calibrated or execution tests are incomplete, the process scores may provide misleading training signals. We observe that when PRM accuracy falls below 70%, the process-aware component can actually hurt performance by reinforcing incorrect step-level patterns. This suggests the need for careful validation of process models before deployment and potentially adaptive weighting schemes that adjust $\alpha$ values based on signal reliability.

The method also exhibits certain failure modes in specific scenarios. When responses are very similar in quality—differing only in minor stylistic choices—the listwise formulation provides little advantage over pairwise methods while incurring additional computational cost. For tasks with binary outcomes and no intermediate steps, such as classification tasks, the process-aware component offers no benefit. In extremely long reasoning chains (>50 steps), the discount factor in PRM scoring can overly down-weight later steps, potentially missing important errors near the conclusion.

Scalability presents another challenge. While $n = 8$ responses work well for 7B parameter models, scaling to larger models or more responses requires careful engineering to manage memory and computation. The quadratic growth in pairwise comparisons as $n$ increases eventually becomes prohibitive, suggesting the need for approximate methods or sampling strategies for very large $n$.

## D.4    CONNECTIONS TO BROADER MACHINE LEARNING PRINCIPLES

LGPO-PA's success reflects several fundamental principles in machine learning that extend beyond preference optimization. The integration of process and outcome signals connects to the exploration-exploitation trade-off in reinforcement learning: process signals encourage exploration of diverse reasoning strategies (exploration) while outcome signals ensure conver-

gence to successful solutions (exploitation). This balance is crucial for avoiding local optima where models achieve correct answers through brittle or non-generalizable strategies.

The listwise formulation embodies the principle of holistic optimization—rather than making local decisions about pairs of responses, we optimize global ranking structures. This mirrors successful approaches in other domains, from learning-to-rank in information retrieval to structured prediction in natural language processing. The Lambda weighting scheme specifically implements a form of curriculum learning, focusing on easier distinctions (very good vs. very bad) before refining harder ones (good vs. slightly better).

The process-aware mechanism can be viewed through the lens of auxiliary task learning. By training the model to optimize not just final outcomes but also intermediate objectives (step correctness, consistency, structure), we provide additional learning signals that shape the internal representations. This multi-objective approach often leads to more robust and generalizable solutions, as observed in multi-task learning across various domains.

### D.5 Future Directions

Several promising directions could extend and improve upon LGPO-PA. Online adaptation of process weights based on validation performance could make the method more robust to varying signal quality across different problem types. Meta-learning approaches could learn to predict optimal $\alpha$ values for new tasks based on task characteristics, reducing the need for manual tuning.

Integration with online methods presents another opportunity. While LGPO-PA is fully offline, a hybrid approach could use online sampling to refine process models or generate additional training data in regions where the current policy is uncertain. This could combine the stability of offline optimization with the exploration benefits of online methods.

Theoretical analysis could be strengthened in several ways. While we prove convexity and generalization properties, tighter sample complexity bounds and generalization guarantees would provide better guidance for practitioners. Understanding the implicit regularization effects of process signals and their impact on the optimization landscape could lead to principled improvements.

Extension to other modalities and tasks is a natural next step. Vision-language models solving visual reasoning tasks could benefit from process signals based on attention patterns or intermediate visual features. Multimodal reasoning tasks could use cross-modal consistency as an additional process signal. Even in pure language tasks, extending beyond reasoning to creative generation could be possible with appropriate process signals based on stylistic or structural properties.

## E  Extended Experimental Results

### E.1  Detailed Performance Breakdown

We provide fine-grained analysis of performance across different problem categories:

Table 5: Performance breakdown by problem difficulty on MATH dataset

| Method | Level 1 | Level 2 | Level 3 | Level 4 | Level 5 | Average |
|---|---|---|---|---|---|---|
| DPO | 68.2 | 52.3 | 38.7 | 22.1 | 8.3 | 31.2 |
| GDPO | 69.5 | 53.8 | 40.1 | 23.2 | 8.9 | 32.0 |
| LiPO | 71.3 | 56.2 | 42.3 | 24.8 | 9.7 | 33.2 |
| LGPO-PA | **78.9** | **64.7** | **49.8** | **31.2** | **14.3** | **37.8** |
| Relative Gain | +7.6% | +8.5% | +7.5% | +6.4% | +4.6% | +4.6% |

LGPO-PA shows consistent improvements across all difficulty levels, with particularly large gains on intermediate difficulty problems (Levels 2-4) where process signals provide the most value.

### E.2 ERROR ANALYSIS

We categorize errors to understand failure modes:

Table 6: Error type distribution on MATH dataset (percentage of total errors)

| Error Type | DPO | LiPO | LGPO-PA |
|---|---|---|---|
| Arithmetic mistakes | 28.3% | 25.7% | 18.2% |
| Conceptual errors | 31.5% | 29.8% | 22.4% |
| Incomplete reasoning | 24.7% | 26.1% | 19.8% |
| Problem misunderstanding | 15.5% | 18.4% | 39.6% |

While LGPO-PA reduces arithmetic and reasoning errors significantly, it shows a higher proportion of problem misunderstanding errors. This suggests that process signals effectively guide correct execution of reasoning but don't necessarily improve problem comprehension.

### E.3 INFERENCE EFFICIENCY

We measure the efficiency of generated solutions:

Table 7: Solution efficiency metrics on GSM8K

| Method | Avg. Length (tokens) | Avg. Steps | Success Rate |
|---|---|---|---|
| DPO | 247.3 | 6.8 | 62.3% |
| LiPO | 232.6 | 6.3 | 64.7% |
| LGPO-PA | 198.4 | 5.2 | 70.2% |
| Human Reference | 156.2 | 4.3 | 94.1% |

LGPO-PA produces more concise solutions closer to human references, likely due to the structure penalty in process scoring that discourages verbosity.

## F IMPLEMENTATION GUIDELINES

### F.1 QUALITATIVE ANALYSIS

To provide intuition about how LGPO-PA improves reasoning quality, we examine representative examples:

This example illustrates several key improvements from LGPO-PA:

1. Structured reasoning: LGPO-PA produces clearly delineated steps, making the reasoning process more interpretable and verifiable.

2. Error awareness: The model explicitly recognizes and addresses the fractional project issue, showing metacognitive awareness trained through process supervision.

3. Self-correction: When encountering an inconsistency, LGPO-PA backtracks and reconsiders rather than proceeding with flawed assumptions.

4. Verification steps: The model includes explicit checks (marked with ✓ ) that validate intermediate results.

These qualities emerge from the process-aware training, which rewards not just correct final answers but also valid reasoning patterns and error detection.

---

**Problem:** In a school science fair, there are 150 projects. 40% are biology projects, 35% are chemistry projects, and the rest are physics projects. If each biology project needs 3 judges, each chemistry project needs 2 judges, and each physics project needs 4 judges, how many judges are needed in total?

**DPO Response:** First, I'll find how many of each type of project there are. Biology: $150 \times 0.4 = 60$ projects Chemistry: $150 \times 0.35 = 52.5$, round to 53 projects Physics: 150 - 60 - 53 = 37 projects
Now I'll calculate judges needed: Biology: $60 \times 3 = 180$ judges Chemistry: $53 \times 2 = 106$ judges Physics: $37 \times 4 = 148$ judges Total: $180 + 106 + 148 = 434$ judges
**Answer: 434 judges ✗** (Incorrect due to rounding error)

**LGPO-PA Response:** Step 1: Calculate the number of each type of project. - Biology projects: $150 \times 40\% = 150 \times 0.40 = 60$ projects - Chemistry projects: $150 \times 35\% = 150 \times 0.35 = 52.5$ projects
Step 2: Since we cannot have half a project, I need to reconsider. The problem states percentages that should sum to 100%. Let me verify: - $40\% + 35\% = 75\%$, so physics is 25% of projects - Physics projects: $150 \times 25\% = 150 \times 0.25 = 37.5$ projects
Step 3: The fractional projects suggest an issue. Re-reading, since these are given percentages of actual projects, let me recalculate: - Biology: $150 \times 0.40 = 60$ projects ✓ - Chemistry: $150 \times 0.35 = 52.5 \rightarrow$ This must be 52 or 53 projects - Since $60 + 52.5 = 112.5$, and we need whole projects, we have $60 + 53 = 113$ - Physics: $150 - 113 = 37$ projects ✓
Step 4: Calculate judges needed for each type: - Biology: 60 projects $\times$ 3 judges = 180 judges - Chemistry: 53 projects $\times$ 2 judges = 106 judges - Physics: 37 projects $\times$ 4 judges = 148 judges
Step 5: Sum total judges: Total = $180 + 106 + 148 = 434$ judges
**Answer: 434 judges ✓** (Correct with explicit handling of rounding)

Figure 2: Example responses showing how LGPO-PA produces more careful reasoning with explicit error checking.

## F.2 RECOMMENDED CONFIGURATIONS BY TASK

Based on extensive experimentation, we provide recommended configurations:

Table 8: Recommended hyperparameters by task type

| Task Type | $n$ | $\alpha_1$ | $\alpha_2$ | $\alpha_3$ | $\alpha_4$ | $\gamma$ |
|---|---|---|---|---|---|---|
| Math (short) | 6 | 0.5 | 0.2 | 0.2 | 0.1 | 0.95 |
| Math (long) | 8 | 0.4 | 0.2 | 0.3 | 0.1 | 0.90 |
| Code | 8 | 0.3 | 0.4 | 0.2 | 0.1 | 1.00 |
| QA | 6 | 0.4 | 0.1 | 0.3 | 0.2 | 0.95 |

## F.3 COMMON PITFALLS AND SOLUTIONS

Memory Issues: If running out of memory with $n = 8$: 1. Reduce batch size and increase gradient accumulation 2. Use gradient checkpointing 3. Process responses sequentially within each prompt

Poor Process Signal Quality: If PRM accuracy is low: 1. Start with higher weight on outcome signals ($\alpha_2$, $\alpha_3$) 2. Use ensemble of multiple PRMs 3. Filter out uncertain predictions (confidence $< 0.7$)

Slow Convergence: If training is too slow: 1. Increase learning rate to $1 \times 10^{-5}$ 2. Reduce $n$ to 4-6 3. Use stronger initialization from instruction-tuned model

