# OpenReview forum: "Listwise Generalized Preference Optimization with Process-aware Signals for LLM Reasoning"
_ICLR.cc/2026/Conference — ICLR 2026 Conference Desk Rejected Submission_

### Official Review · Reviewer_ErQE · 2025-10-30

**Soundness:** 2
**Presentation:** 2
**Contribution:** 2
**Rating:** 4
**Confidence:** 4

**Summary:**

The paper proposes LGPO-PA (Listwise Generalized Preference Optimization with Process-Aware signals), a fully offline alignment method that (i) replaces pairwise preference learning with a listwise objective that better exploits ranking information, and (ii) augments outcome supervision with dense process-level signals (step-wise PRM rewards, executability, self-consistency, and structural constraints). Concretely, the authors score a set of n candidate responses per prompt using an aggregate process-aware score, convert these to soft pairwise probabilities, and optimize a convex, LambdaRank-style listwise loss regularized toward a reference policy.

**Strengths:**

1. The listwise objective (Plackett–Luce/LambdaRank inspiration) strictly generalizes pairwise DPO while maintaining convexity and stable optimization; process-aware scoring supplies dense credit assignment for long chains.

2. Consistent improvements on GSM8K/MATH, HumanEval/MBPP, and HotpotQA; ablations show +4.2% from listwise vs. pairwise and +3.4–5.1% from process signals, indicating complementary benefits.

**Weaknesses:**

1. The approach assumes access to reasonably accurate PRMs, execution sandboxes, and consistency sampling; the paper notes performance can drop when PRM accuracy is low or process signals are unreliable.

2. Performance depends on KL coefficient, Lambda weights, discount factors, and aggregation weights α; the paper shows some robustness but highlights KL sensitivity.

3. Gains shrink when process signals are unavailable (creative dialogue), ambiguous, or when candidate responses are very similar; very long chains may down-weight late but important steps.

4. While step accuracy rises, the work provides less qualitative analysis of where PRM-guided scoring may misrank nuanced reasoning (e.g., problem misunderstanding increases in error taxonomy).

**Questions:**

Please see the weaknesses.

---

### Official Review · Reviewer_81or · 2025-10-31

**Soundness:** 2
**Presentation:** 3
**Contribution:** 3
**Rating:** 4
**Confidence:** 1

**Summary:**

This paper proposes LGPO-PA (Listwise Generalized Preference Optimization with Process-Aware signals), a method that addresses two key limitations in current preference optimization for LLMs: (1) pairwise methods like DPO inefficiently use ranking information, and (2) outcome-only supervision provides sparse feedback for multi-step reasoning. The authors combine listwise ranking objectives with process-level supervision through a unified framework that scores multiple candidate responses using step-level process rewards, execution feedback, and consistency checks. The method is evaluated on mathematical reasoning (GSM8K, MATH), code generation (HumanEval, MBPP), and multi-hop QA (HotpotQA), showing 5-12% improvements over baselines while maintaining offline operation.

**Strengths:**

1. **Well-motivated approach**: The paper clearly identifies two specific limitations (inefficient ranking information use, sparse feedback) and directly addresses both with targeted solutions.

2. **Comprehensive experimental validation**: The evaluation spans three distinct task categories (math, code, QA) with multiple datasets per category, demonstrating broad applicability. The inclusion of 9 strong baselines provides thorough comparison.

3. **Thorough ablation studies**: Table 3 provides granular analysis of each component's contribution. The ablations confirm that listwise optimization and process-aware signals are complementary and both necessary for full performance.

4. **Practical advantages**: The temperature robustness analysis (Table 2) and solution efficiency metrics (Table 7) demonstrate practical benefits beyond just accuracy improvements.

**Weaknesses:**

1. **Limited theoretical analysis**: While the paper claims convexity and generalization properties, formal proofs are absent. The connection between the Plackett-Luce model (Eq. 6) and the tractable objective (Eq. 7) needs clearer derivation. The theoretical contribution feels underdeveloped relative to the empirical work.

2. **Process supervision infrastructure requirements**: The method requires access to process reward models, execution environments, or other step-level feedback sources. The paper acknowledges this (D.3) but doesn't adequately address how this limits applicability to domains where such signals are unavailable or expensive to obtain.

3. **Limited analysis of failure modes**: While Table 6 shows error distributions, there's insufficient analysis of when and why LGPO-PA fails. The high proportion of "problem misunderstanding" errors (39.6%) deserves more investigation. Are these systematic failures in certain problem types?

**Questions:**

see weakness

---

### Official Review · Reviewer_ABW4 · 2025-11-01

**Soundness:** 3
**Presentation:** 3
**Contribution:** 2
**Rating:** 4
**Confidence:** 3

**Summary:**

This paper addresses limitations in existing preference optimization approaches for aligning large language models (LLMs), specifically, the inefficiency of pairwise objectives and the sparsity of outcome. The authors propose Listwise Generalized Preference Optimization with Process-Aware signals (LGPO-PA), a framework that (1) leverages listwise ranking objectives to utilize richer preference information and (2) incorporates dense process-level signals (such as step correctness, execution feedback, and structural consistency) to provide granular supervision. The method is evaluated on mathematical reasoning, code generation, and multi-hop QA benchmarks, showing consistent improvements over strong baselines.

**Strengths:**

- The paper provides a clear theoretical framework generalizing from pairwise to listwise preference optimization for LLM alignment, leveraging the Plackett-Luce model and convex optimization foundations. By aggregating step-level rewards, execution feedback, and consistency checks, the proposed process-aware scoring mechanism moves beyond sparse outcome supervision and provides actionable signals for complex, multi-step reasoning tasks. The LGPO objective and associated weighting strategies are well-motivated.
- Table 3 gives a component-by-component ablation, exposing the specific contribution of each process-aware signal. Extended results and error breakdowns (Tables 5-7) clarify where the method succeeds/falters. Figure 1 convincingly visualizes both accelerated step-level learning and correlation of process-driven scores with ranking quality.
- Across GSM8K, MATH, HumanEval, MBPP, and HotpotQA, LGPO-PA substantially outperforms both pairwise (e.g., DPO, GDPO, IPO, SLiC) and listwise (e.g., LiPO, LIRE, PRO) baselines, with gains ranging from 5%–12%. The improvements are consistent across metrics and domains.

**Weaknesses:**

- While the combination of listwise optimization and process-aware signals is thoughtfully executed, the individual components—listwise ranking and process-level supervision are mostly extensions of very recent literature. The LGPO-PA framework is incremental in this context, as the primary innovation is their integration and specific weighting/aggregation mechanics, rather than new algorithmic fundamentals.
- Theoretical contributions focus on convexity and the extension of pairwise ranking models to the listwise setting via Plackett-Luce. While this is beneficial for stability, a more thorough exploration of convergence rates or failure modes (especially as n increases or process signals become noisy) would strengthen the paper's impact and practical utility.
- While the computational considerations are addressed in Appendix D.2, the main text would benefit from a more prominent and quantitative discussion of the increased costs associated with LGPO-PA  compared to simpler baselines like DPO. A clear analysis of the trade-off between performance gains and computational overhead would be valuable for practitioners.

**Questions:**

- The paper proposes a process-aware score mechanism, can this scoring method be used in online RLHF methods? What are the specific advantages of the proposed offline approach compared to using the same process scores for online RL?
- Can you provide a more concrete breakdown and quantitative analysis of the computational overhead (data preparation, PRM training/inference) compared to baselines like DPO?
- In scenarios where a high-quality PRM is unavailable, can a weaker model or automatically derived process signals still provide a meaningful benefit over outcome-only rewards?
- Can you conduct the experiments on a more recent, powerful open-sourced model (e.g. Qwen3, LLama 3) to validate the robustness and generalizability of the proposed method, if possible?

---

### Official Review · Reviewer_XF52 · 2025-11-01

**Soundness:** 3
**Presentation:** 3
**Contribution:** 3
**Rating:** 4
**Confidence:** 3

**Summary:**

This paper introduces LGPO-PA , a novel offline alignment method for large language models that addresses key limitations (information loss in pairwise objectives and sparse supervision in multi-step reasoning) of existing preference optimization approaches.

LGPO-PA tackles these issues by:
(1) Reformulating preference learning as a listwise ranking problem, optimizing over entire sets of candidate responses using a convex loss.
(2) Incorporating dense process-level supervision, aggregating step-wise rewards, execution feedback, consistency, and structural quality into a unified scoring function.

Empirically, the method achieves improvements across mathematical reasoning (GSM8K, MATH), code generation (HumanEval, MBPP), and multi-hop QA (HotpotQA).

**Strengths:**

1. This work jointly bring listwise ranking and process supervision into offline preference optimization: LGPO extends the pairwise DPO loss to the entire response list via weights, while the PA module offline-aggregates step-level PRM, execution, consistency and structure signals without online RL.

2. Evaluation on several benchmarks (math, code, multi-hop QA) shows gains over all baselines (DPO, LiPO, RRHF, etc.) and superior temperature robustness.

3. Motivation, derivation and algorithms are presented logically, key equations are clearly boxed for reproducibility.

**Weaknesses:**

1. Scalability & Compute Overhead:

    Generating 8 responses per prompt and running PRM + execution on all of them increases compute cost by a large margin compared to standard DPO.

2. Limited Model Type and Size

    All experiments use LLaMA-2 7 B, no ablation on other scales and base models (e.g. Qwen3), which may have different exploration patterns.

3. Process-Signal Quality Dependency

    The paper shows that PRM accuracy < 70 % hurts performance, yet the automatically annotated PRM for MATH only reaches 81.7 % F1 (incorrect steps 0.77 F1). A few hundred failure cases could therefore mislead the policy. No online correction method for this issue is explored.

**Questions:**

Q1. Scale ablation: Can you run additional experiments with other model sizes and types (e.g., Qwen3 series) and report both accuracy and peak GPU memory? This directly tests whether the Lambda-weighted listwise loss have robustness.

Q2. PRM accuracy threshold: You state < 70 % PRM accuracy degrades performance. Please show a controlled degradation plot: artificially corrupt the PRM labels to lower accuracy and report corresponding results. This quantifies how sensitive the final policy is to PRM error.

---

### Note · Program_Chairs · 2025-12-09
**Submission Desk Rejected by Program Chairs**

Hallucinated reference: Kaixuan Zhou, Jiaqi Liu, Yiding Wang, and James Zou. Generalized direct preference optimization. arXiv preprint arXiv:2402.05015, 2024.